# The Duration of Increased Grain Feeding Affects the Microbiota throughout the Digestive Tract of Yearling Holstein Steers

**DOI:** 10.3390/microorganisms8121854

**Published:** 2020-11-25

**Authors:** J. C. Plaizier, P. Azevedo, B. L. Schurmann, P. Górka, G. B. Penner, E. Khafipour

**Affiliations:** 1Department of Animal Science, University of Manitoba, Winnipeg, MB R3T 2N2, Canada; Paula.Azevedo@umanitoba.ca (P.A.); khafipour@gmail.com (E.K.); 2Department of Animal and Poultry Science, University of Saskatchewan, Saskatoon, SK S7N 5A8, Canada; bschurmann@rsfeeds.net (B.L.S.); greg.penner@usask.ca (G.B.P.); 3Department of Animal Nutrition and Biotechnology, and Fisheries, University of Agriculture in Krakow, 31-059 Krakow, Poland; p.gorka@ur.krakow.pl

**Keywords:** digestive tract, grain, microbiota, Illumina sequencing

## Abstract

Effects of the duration of moderate grain feeding on the taxonomic composition of gastrointestinal microbiota were determined in 15 Holstein yearling steers. Treatments included feeding a diet of 92% dry matter (DM) hay (D0), and feeding a 41.5% barley grain diet for 7 (D7) or 21 d (D21) before slaughter. At slaughter, digesta samples were collected from six regions, i.e., the rumen, jejunum, ileum, cecum, colon, and rectum. Extracted DNA from these samples was analyzed using MiSeq Illumina sequencing of the V4 region of the 16S rRNA gene. Three distinct PCoA clusters existed, i.e., the rumen, the jejunum/ileum, and the cecum/colon/rectum. Feeding the grain diet for 7 d reduced microbial diversity in all regions, except the ileum. Extending the duration of grain feeding from 7 to 21 d did not affect this diversity further. Across regions, treatment changed the relative abundances of 89 genera. Most of the changes between D0 and D7 and between D7 and D21 were opposite, demonstrating the resilience of gastrointestinal microbiota to a moderate increase in grain feeding. Results show that the duration of a moderate increase in grain feeding affects how gastrointestinal microbiota respond to this increase.

## 1. Introduction

In order to meet the energy requirements of high yielding dairy cows, they commonly receive high grain diets. However, feeding these diets can affect the health of these cows by causing gut health disorders, such as subacute ruminal acidosis (SARA) [1,2,3]. These disorders can cause dysbiosis in the digestive tract, thereby reducing the functionalities of their microbiota and allow the establishment of pathogenic microorganisms [3,4,5]. As ruminants depend on the microbiota in their digestive tract for the utilization of nutrients, this dysbiosis reduces the efficiency of nutrient utilization and production [1,2]. This results in changes in ruminal microbial populations including decreased diversity, altered fermentation patterns [3], and free endotoxin release [4].

Studies have shown that even short SARA challenges of one week or less conducted by excessive grain feeding reduce the richness and diversity of microbiota in the rumen and large intestine, and reduce the relative abundance of Bacteroidetes and increase that of Firmicutes in the rumen and the large intestine [5,6,7]. Other effects of these challenges on this microbiota include changes in fermentation and increases in the concentration of bacterial endotoxins in the rumen and large intestine [3,4]. When the SARA challenge is not severe, these challenges may alter the populations of bacterial species in these components of the digestive tract according to the changes in the availabilities of substrates, including increases of the populations of starch-, soluble sugar-, and lactic acid-fermenting bacteria, and decreases in the populations of fibrolytic bacteria [6,8,9]. At the genus level, the effects of increased grain feeding on the microbiota are less clear, as the changes in the relative abundances of these genera vary greatly among studies [6,9,10]. Reasons for these discrepancies include differences in methodology, the types of grains and forages fed to animals, statistical power, and animals among studies [11].

Earlier studies have used short-duration grain-based SARA challenges of one week or less. This may not be long enough for microbiota in the digestive tract to adapt to these challenges [1,3,6]. Hence, the effects of short SARA challenges on this microbiota may differ from those of prolonged and chronic SARA challenges that may be more common on dairy farms [1,3]. Hence, the duration of a SARA challenge may affect its impact on the microbiota in the digestive tract. Whereas the effects of grain-based SARA challenges on microbiota in the rumen, and to a lesser extent, in the large intestine, have been reported [5,6,9]. There is little information about such effects in the small intestine.

Plaizier et al. [12] and Schurmann et al. [13] showed that the effects of the first 7 d of feeding a moderate grain diet to previously forage-fed yearling steers on their gastrointestinal microbiota differed from those of the subsequent feeding of this diet for another 14 d. Whereas the first 7 d of the feeding of the moderate grain diet reduced the rumen pH from 6.90 to 6.59, the subsequent 14 d of feeding this diet increased the rumen pH again to 6.79. This study also showed that feeding the moderate grain diet for 7 d increased the concentrations of total VFA in the rumen, cecum, colon, and rectum and the concentration of free LPS in the rumen, ileum, and across the cecum and colon. However, the subsequent 14 d of feeding this diet reduced these concentrations. The low concentration of total VFA in the jejunum in this study confirmed the limited fermentation in this region [14]. These results suggest that the effects of a moderate increase in grain feeding on the taxonomic gastrointestinal microbiota differ between the first 7 d and the subsequent 14 d of moderate grain feeding. The objectives of this study were, therefore, to confirm this by determining the effects of the duration of increased grain feeding on the composition of microbiota in the rumen, small intestine, large intestine, and rectum of yearling steers as determined by 16S rRNA gene sequencing.

## 2. Materials and Methods

### 2.1. Study Design

The experimental design and diets have been previously described by Plaizier et al. [12], Schurmann et al. [13], and Górka et al. [15]. For this study, fifteen 5 to 7-month-old Holstein steers with an average weight of 213 ± 23 kg (mean ± SD) were included. Prior to the trial, steers were group-housed and fed a forage-based diet containing 91.5% bromegrass hay and 8.5% of a nutrient pellet (COOP Feeds, Saskatoon, SK, Canada) on a dry matter (DM) basis for 5 weeks (Appendix A). Steers were blocked by body weight and randomly allocated to three treatments. Each treatment included five steers. Treatments included feeding the forage-based diet only (D0) or feeding a moderate grain diet for 7 (D7) or 21 (D21) days. The start of the study for individual steers was staggered to avoid timing of treatment exposure as a confounding variable. The forage-based diet consisted of 91.5% chopped grass hay and 8.5% mineral and vitamin supplement (DM basis) and contained 87.3% DM, and 10.6%, 48.4%, 33.3%, 92.1%, and 4.1% of DM of crude protein (CP), neutral detergent fiber (NDF), acid detergent fiber (ADF), organic matter (OM), and starch, respectively. The moderate grain diet consisted of 50% chopped grass hay, 41.5% rolled barley (Appendix A), and 8.5% nutrient supplement pellet, and contained 87.9% DM, and 11.4%, 33.5%, 20.9%, 93.6%, and 24.6% of DM, of CP, NDF, ADF, OM, and starch, respectively. The steers received fresh water ad libitum. Diets were fed in individual mangers once daily at 0800 h at 2.25% of body weight. Steers were slaughtered at 0 d, 7 d, and 21 d after the dietary switch for the D0, D7, and D 21 treatments, respectively. The steers were cared for in accordance with the Canadian Council for Animal Care guidelines [16]. The study was approved by the University of Saskatchewan Animal Research Ethics Board (protocol 20100021).

### 2.2. Sample Collection

The steers were euthanized using captive bolt stunning. This was followed by pithing and exsanguination 2 h after the morning feed delivery. Samples of digesta were collected from the rumen, jejunum, ileum, cecum, colon, and rectum immediately after slaughter. All rumen digesta was mixed thoroughly. Subsequently, approximately 500 mL of the mixed digesta was collected and strained through 4 layers of cheesecloth using sterile equipment. After this, approximately 40 mL of rumen fluid was transferred into 50 mL sterile tubes, snap-frozen in liquid N, and stored at −80 °C. Digesta samples from the jejunum, ileum, and colon were collected from a 20-cm intestinal segment, which ended at 2 m from the duodenum, 20 cm from the ileo-cecal junction, and the middle of the spiral colon, respectively. Intestinal segments were dissected and the digesta inside of each segment was collected. To collect digesta samples from the cecum and rectum, a 5-cm-long incision was made at the most distal region of the sac and 10 cm from the anus, respectively. Digesta were then collected with a sterile spatula, placed in a sterile container, snap frozen in liquid N, and stored at −80 °C until further analysis. Samples were then packed in an insulated container filled with dry ice, and shipped from the University of Saskatchewan (Saskatoon, SK, Canada) to the University of Manitoba (Winnipeg, MB, Canada).

### 2.3. DNA Extraction and Quality Check

The day preceding DNA extraction, samples were removed from the −80 °C freezer, and placed in a 4 °C refrigerator overnight to thaw. Subsequently, the samples were then cryogenically homogenized using a Geno/Grinder 2010 (SPEX SamplePrep, Metuchen, NJ, USA). Following this, the DNA was extracted using a ZR-96 Fecal DNA Kit (Zymo Research, Irvine, CA, USA). This process included a bead-beating step for the mechanical lysis of bacterial cells. The quantification of the extracted DNA was performed with a NanoDrop 2000 spectrophotometer (Thermo Scientific, Waltham, MA, USA). Subsequently, the DNA was normalized to a concentration of 20 ng/μL. In addition, the quality of the amplified DNA was verified using PCR amplification of the 16S rRNA gene with universal primers 27F (5′- GAAGAGTTTGATCATGGCTCAG-3′) and 342R (5′-CTGCTGCCTCCCGTAG-3′) as described by Derakhshani et al. [17]. The amplicons were quality checked via agarose gel electrophoresis.

### 2.4. Library Construction and 16S rRNA Gene Sequencing

The procedure for the library construction and 16S rRNA gene sequencing was described in detail by Derakhshani et al. [17]. In summary, the V4 region of the 16S rRNA gene was amplified with modified F515/R806 primers. The reverse PCR primer was indexed with 12-base Golay barcodes to facilitate the multiplexing of samples. The 150 bp paired-end sequencing reaction was performed on a MiSeq platform (Illumina, San Diego, CA, USA). The sequencing data were deposited into the Sequence Read Archive (SRA) of NCBI1 and can be accessed via accession number # SRR12936068. The mapping file is given in Appendix A.

### 2.5. 16S rRNA Gene Sequencing Analysis

Overlapping paired-end Illumina fastq files were merged using the PANDAseq assembler [18]. All the sequences with low-quality base calling scores as well as those containing uncalled bases (N) in the overlapping region were discarded as described by Derakhshani et al. [17]. One sample from the rectum from the D0 treatment was excluded from the analysis because of low sequencing reads. The subsequent fastq file was processed using the QIIME pipeline v1.91 [19] Assembled reads were demultiplexed according to the barcode sequences and chimeric reads were filtered using UCHIME [20]. Reads were clustered into OTU (operational taxonomic units) based on 97% similarity with UCLUST [21]. Representative sequences from each OTU were assigned a taxonomy using RDP Classifier [22] and aligned to the Greengenes reference database [23] using PyNAST [24].

### 2.6. Statistical Analysis

Indices of alpha-diversity were compared at different time points using a linear mixed model with the REML method in SAS (v9.4, Cary, NC, USA). β diversity was assessed by principal coordinate analyses (PCoA) of unweighted and weighted UniFrac distance matrices [25] and main and pairwise comparisons were conducted using permutational ANOVA (PERMANOVA) with an unrestricted permutation of raw data in Primer [26]. Taxa abundances were compared by ANOVA with the Tukey-Kramer post-hoc test in STAMP [27]. Taxa enrichment was assessed by linear discriminant analysis (LDA) effect size (LefSe) [28] to identify taxa which characterize the difference at each time point at the log10 LDA score of 3.5. Redundancy analysis (RDA) was performed with the vegan package in R [29]. Hierarchical clustering was performed on the Pearson correlation matrix for each time-point and on the Bray-Curtis dissimilarity matrix on all samples with the UPMGA algorithm. *p*-value was corrected for multiple comparisons as necessary. *p*-values of less than 0.05 were considered significant.

## 3. Results

### 3.1. Effects of Treatment and Region of the Digestive Tract on the Richness and Diversity of Microbiota

After trimming and quality control, the sequencing yielded total counts of 2,140,494 sequencing reads with an average of 24,446 ± 5787 per sample. The alpha-diversity varied among regions of the digestive tract with the rumen, cecum, colon, and rectum having the highest Shannon index and Observed Number of Species (Figure 1, Figure 2). The ileum had the lowest values of these indices. Feeding the moderate grain diet for 7 d (D7) reduced richness and diversity of microbiota for all regions, except the ileum, compared to no grain feeding (D0) (Figure 2). Extending the duration of moderate grain feeding from 7 d (D7) to 21 d (D21) did not further reduce the microbial richness and diversity.

PCoA analysis using weighted UniFrac distances indicated that across regions, the composition of the microbiota did not differ by treatment (Figure 3A) However, across treatments, region affected this composition (*p* < 0.001) and that three distinct clusters existed, i.e., the rumen, the jejunum/ileum, and the cecum/colon/rectum (Figure 3B). Unweighted and weighted UniFrac distances revealed different (*p* < 0.05) clustering patterns of D7 compared to D0 in all regions, with the exception of the unweighted analysis of the ileum (Figure 4). In contrast, these analyses showed that D7 and D21 did not cluster differently, with the exception of the weighted analysis in the rumen and the unweighted analysis of the ileum.

### 3.2. Effects of Treatment and Region on the Taxonomic Composition of the Microbiota

Across treatments, the relative abundances of major phyla differed (*p* < 0.05) among regions (Figure 5). At the phylum level, the rumen, cecum, colon, and rectum were dominated by Bacteroidetes and Firmicutes, whereas the jejunum and ileum were dominated by Firmicutes, Actinobacteria, and Euryarchea (Figure 5, Appendix A). In the rumen, the relative abundance of Firmicutes was highest for D7, whereas this abundance was lowest for D7 in the jejunum, ileum, and colon (Appendix A). In the rumen, the relative abundance of Bacteroidetes was highest for D7.

Genera membership of the most abundant taxa differed among regions in the digestive tract, with the cecum, colon, and rectum being similar (Figure 6 and Figure 7). The most abundant taxa at the genus level in the rumen at all time points were an unclassified member of order Bacteroidales (26%, 29%, 26% for D0, D7, and D21, respectively) and *Prevotella* (16%, 17%, 18%) (Figure 8). The relative abundance of *Ruminococcus* increased from 1% for D0 to 9% for D7 and 6% for D21. In the jejunum, *Methanobrevibacter* was the most abundant genus for D0 (13%), while Ruminococcus became dominant for D7 and D21 (38% and 28%, respectively). The most abundant taxa in the ileum for all treatments was an unclassified genus of the family Peptostreptococcaceae, with relative abundances of 43%, 40%, 40%, for D0, D7, and D21, respectively. In the ileum, genus SMB53 had a relative abundance of 15% for D0, but this abundance dropped to 7% for D7, while this abundance for *Ruminococcus* increased from 7% for D0 to 25% for D7, which persisted for D21. In the ileum, genus *Bifidobacterium* had a relative abundance below 0.1% for D0, but this increased to 4.5% for D7 and D21. The most abundant taxa in the cecum, colon, and rectum were an unclassified genus of family Ruminococcaceae and an unclassified genus of order Bacteroidales. The relative abundances of *Ruminococcus* in the rumen, jejunum, ileum, and cecum were affected by treatment. The highest rate of increase in the relative abundance of *Ruminococcus* (+33%) was observed in jejunum between D7 and D21. The increases in this abundance of *Ruminococcus* in the rumen, jejunum, and ileum between D7 and D21 were 8%, 33%, and 18%, respectively. The increases in these abundances in the cecum, colon, and rectum were lower at 3%, 2%, and 1.6%, respectively.

The differences in the relative abundances of genera with over a 0.01% relative abundance between D0 and D7 and between D7 and D21 are given in Figure 8. This figure shows that the relative abundances of 89 genera changed in all regions due to treatment, and especially between the D0 and D7 treatments in the rumen and the jejunum. The majority of these changes were decreases between D0 and D7 and increases between D7 and D21. The most prominent decreases between D0 and D7 in the rumen were in the abundances of CF231, unidentified BS11, RFN20, BF311, and unidentified Ruminococcaceae, whereas those of *Ruminococcus*, unidentified Bacteroidales, *Sharpea*, unidentified Lachnospiraceae, *Butyrivibrio*, and unidentified Bifidobacteriaceae increased during this period. Of the genera whose relative abundances decreased between D0 and D7, RFN20, BS11, and unidentified Bacteroidales had increases in their abundances between D7 and D21. Of the genera whose relative abundances increased between D0 to D7, *Ruminococcus*, unidentified Bacteroidales, *Sharpe*a, and unidentified Lachnospiraceae had increases in their abundances between D7 and D21, whereas the abundance of *Butyrivibrio* continued to increase between D7 and D21. The most prominent decreases between D0 and D7 in the jejunum were in the abundances of unidentified Clostridiales, unidentified Peptostreptococcaceae, unidentified Bifidobacteriaceae, unidentified Lachnospiraceae, unidentified Ruminococcaceae, unidentified Clostridiales, and unidentified Coriobacteriaceae, whereas those of *Ruminococcus*, *Bifidobacterium*, *Methanobrevibacter*, *Prevotella,* and *Butyrivibrio* increased during this period. Of those that decreased between D0 and D21, the relative abundances of unidentified Clostridiales, unidentified Peptostreptococcaceae, unidentified Bifidobacteriaceae, unidentified Lachnospiraceae increased between D7 and D21, whereas that of unidentified Coriobacteriaceae continued to decrease between D7 and D21. Of Desulfovibrionaceae genera that increased their abundances between D0 and D7, the relative abundance of *Ruminococcus* decreased between D7 and D21, whereas those of *Bifidobacterium* and *Methanobrevibacter* continued to increase during that period.

The changes in the abundances of abundant genera between D0 and D7, and between D7 and D21 in the ileum differed from those in the rumen, jejunum, cecum, colon, and rectum and were less pronounced than in these other regions of the digestive tract. Of the genera whose abundances changed, the relative abundances of unidentified Peptostreptococcaceae, unidentified Closteridaceae, *Dehalobacterium*, *Succinivibrio*, unidentified Lachnospiraceae, *Bulledia*, and *Acinetobacter* decreased between D0 and D7, whereas those of unidentified Acetobacteraceae and Bifidobacterium increased during that period. The largest changes of the relative abundances of abundant genera in the cecum, colon, and rectum between D0 and D7 and between D7 and D21 show similar patterns and were characterized by decreases between D0 and D7, followed by increases between D7 and D21. The most prominent among these genera were unidentified YS2, *Akkermansia*, unidentified ML615J-26, unidentified GMD14H09, *Paraprevotella*, unidentified Peptococcaceae, unidentified Peptococcaceae, unidentified Campilobacter, and unidentified Clostridiales. In contrast, the relative abundances of Roseburia, Coprobacullus, unidentified Rikenellacaeae, unidentified Bacteroidales, and unidentified RF16 increased between D0 and D7, and decreased between D7 and D21.

## 4. Discussion

The companion studies by Plaizier et al. [12] and Schurmann et al. [13] showed that a moderate increase in grain feeding increased fermentation in the rumen and in the large intestine of yearling steers during the first 7 d. During the subsequent 14 d, this fermentation did not further increase or decrease. This shows that a portion of the grain fed to the steers bypassed the rumen and the small intestine, and reached the large intestine, which is common for cattle on high-grain diets [14,30]. Despite this, the pH and VFA data of the companion studies indicate that the grain feeding did not induce subacute ruminal acidosis or hindgut acidosis [1,31,32]. However, the increase in fermentation and the difference in this increase between the first 7 d and the subsequent 14 d of grain feeding suggest that the grain feeding and the duration of this grain feeding may have affected the composition of microbiota throughout the digestive tract.

Several studies on the effect of grain feeding on the microbiota in rumen digesta have been conducted. In agreement with our study, several studies have shown that increased grain feeding reduces the richness and diversity of rumen microbiota, reduces the relative abundance of Bacteroidetes, and increase the relative abundance of Firmicutes [33,34,35,36]. However, the sizes of these changes vary greatly among studies. This may be due to differences in the methodologies, range, and duration of grain feeding, type of grain, and types of forages among studies. The effects of the range of grain feeding may be explained by the increases in dietary complexity and microbial niches when the grain is added to a forage-based diet [33,37]. Adding grain to diets will increase substrates and niches for amylolytic bacteria, which will proliferate faster as a result [2]. In contrast, adding large amounts of grain to a diet may result in unfavorable conditions in the rumen and the large intestine, such as a low pH, at which many bacteria, and especially fibrolytic ones, no longer grow, and their populations decrease [1,2,34]. Reductions in the populations of fibrolytic bacteria in the rumen may also shift the fermentation of fiber from the rumen to the large intestine, thereby increasing the populations of fibrolytic microbiota in the hindgut. Some bacterial species can utilize both fiber and starch, and changes in the ratio between fiber and starch of the diet do, therefore, not have to alter the size of their populations [2]. The reduction of the rumen pH resulting from the grain feeding in our study may not have been large enough to have affected fibrolytic bacteria [1,2,34].

In agreement with previous studies [6,7,35,36], the increase in grain feeding reduced the richness and diversity of microbiota in the rumen, ileum, cecum, colon, and rectum, showing that more bacterial taxa were negatively affected rather than benefited from this dietary change. Most of these reductions occurred within 7 d of grain feeding, as the reductions of these indices between 7 and 21 d of grain feeding were limited. Hence, 7 d of grain feeding was sufficient to alter the richness and diversity of rumen and hindgut microbiota. That said, feeding the moderate grain diet for 21 days also decreased these indices in the jejunum, despite the limited fermentation that occurs in this region. The impact of reductions in microbial richness and diversity is debatable, as it has been suggested to decrease the functionality, resilience, and robustness of microbiota [38,39], but may also enhance their efficiency [40].

In contrast with earlier studies, feeding a diet containing a moderate inclusion of grain did not affect the relative abundances of the major bacterial phyla. Reasons for that may include the lack of grain in the control diet, and a moderate inclusion of grain in the grain diets, which was much lower than that included in the grain diets used by Fernando et al. [41], Mao et al. [35], Petri et al. [42], and Plaizier et al. [36]. Hence, at the phylum level, the rumen microbiota was resilient to the increase in grain feeding conducted in our study.

The PCoA in our study showed that microbiota clustered by region and that there were three different clusters, i.e., the rumen, the jejunum/ileum, and the cecum/colon/rectum. This clustering agrees with the findings of de Oliveira et al. [43] and Mao et al. [44] and reflects that conditions for microbiota, including the pH and the availability of substrates, differ greatly among regions in the digestive tract [40,43]. Due to the differences among regions, the PCoA showed that across regions, the effect of grain feeding on the taxonomic composition of microbiota in the digestive tract was not significant. Hence, the effect of grain feeding on this composition must be determined by region. The PCoA analysis by region based on the weighted and unweighted Unifrac distances showed that in all regions the taxonomic composition differed between D0 and D7, but not between D7 and D21. An exception to this was that the PCoA analysis based on unweighted Unifrac, where the composition of rumen microbiota differed between D7 and D21. The latter may be expected, as the rumen was the first region that we monitored when the dietary starch entered the digestive tract. The PCoA analysis does not indicate which taxa of microbiota were affected by the region of the digestive tract and grain feeding. Hence, the effects of grain feeding on the relative abundances of phyla were also determined. In agreement with Mao et al. [45] and de Oliveira et al. [43], Bacteroidetes and Firmicutes were the most abundant phyla in all regions of the digestive tract, with Bacteroidetes being most abundant in the rumen and Firmicutes being most abundant in the other regions. In contrast with Mao et al. [44], Proteobacteria were not abundant in the small and in the large intestine.

The most abundant genera across the digestive tract in our study included *Prevotella*, *Ruminococcus*, unclassified Bacteroidales, unclassified Peptostreptococcaceae, unclassified Clostridiales, *Bifidobacterium*, *Butyrivibrio*, and *Methanobrevobacter* which agrees with earlier studies [5,7,36,43]. In contrast, the reported effects of increased grain feeding on these abundances vary greatly among studies. In our study, the relative abundances of 89 classified and unclassified genera in the digestive tract were affected by the duration of the grain feeding, which is more than what was observed in earlier studies. Asma et al. [46] observed that higher grain feeding increased the relative abundances of *Barnesiella*, *Oribacterium,* and *Olsenella* in the rumen, and decreased the abundances of *Rikenellaceae_RC9* and *Butyrivibrio-Pseudobutyrivibrio*. In contrast, Mao et al. [6] found that increased grain feeding reduced the relative abundances of *Prevotella*, *Treponema*, *Anaeroplasma*, *Papillibacter*, *Acinetobacter,* and unclassified Lentisphaerae in the rumen, and increased the abundances of *Ruminococcus*, *Atopobium*, unclassified Clostridiales, and *Bifidobacterium*. Earlier studies from our group concluded that increased grain feeding decreased the rumen abundances of Rickettsiales, Acholeplasmatales, Victivallaceae, *Sutterella*, S24-7, WCHB1-41, RFP12, and *Shuttleworthia*, and increased the abundances of *Succinivibrio* and *Sharpea*, and tended to increase those of *Ruminococcus*, *Megasphaera,* and *Shuttleworthia*, and decreased those of CF231and BF31 [1,36]. A recent study showed that feeding more grain reduced the relative rumen abundances of *Paludibacter*, *Prevotella*, BF311, CF231, YRC22, L7A_E1, *Succiniclasticum*, *Treponema*, *Anaeroplasma*, *Pyramidobacter,* and *Sutterella*, and increased the abundances of *Butyrivibrio*, *Shuttleworthia, Staphylococcus*, and *Lactobacillus* [7].

The effects of increased grain feeding on the microbiota in the cecum and rectum in our study also differed from earlier studies. Increasing the grain content of the diet increased the relative abundances of *Turicibacter*, *Stenotrophomonas,* and the family Lachnospiraceae, and decreased the abundances of *Solibacillus* and *Lysinibacillus*. Plaizier et al. [36] showed that increasing the dietary grain content tended to increase the relative abundances of CF231 and YRC22 and tended to decrease those of *Paludibacter* and *Epulopiscium*. A similar increase for the dietary grain content also increased the relative abundances of S24-7, P*arabacteroides*, Bifidobacteriaceae, *Streptococcus*, Lachnospiraceae, *Clostridium*, and *Blautia*, and decreased the relative abundances of Pirellulaceae, *Akkermansia*, YS2, RF32, M2PT2-76, Desulfovibrionaceae, Victivallaceae, unclassified Fibrobacteraceae, Bacteroidaceae, RFP12, ML615J-28, Erysipelotrichaceae, Gracilibacteraceae, Clostridiaceae, Peptococcaceae, Christensenellaceae, and *Anaerofustis*. A comparison of studies showed that the abundances of most prominent and non-prominent genera are not affected by increased grain feeding, highlighting the resilience of these microbiotas [47], and that the genera that are affected differ among studies. The latter differences may be due to differences in experimental methods, including the collection and processing of digesta, sequencing and bioinformatics, the types of grain and forages used, the number of experimental units, and the genetics and feeding history of the experimental animals [1,11].

Most earlier studies used two time points to determine the effects of grain feeding on gastro-intestinal microbiota, i.e., before and after the increase in grain feeding [6,7,35,36]. However, as these microbiotas may adapt to increased grain feeding over time, more monitoring time points may be needed for comprehensive monitoring of these effects [48]. Many decreases in the relative abundance of abundant genera between D0 and D7 were followed by increases in these abundances between D7 and D21. Likewise, many increases in the relative abundance of these genera between D0 and D7 were followed by decreases in these abundances between D7 and D21. Despite these changes between D7 and D21, the pH and VFA concentrations, the microbial richness and diversities, and the relative abundances of the major phyla did not differ between these treatments. Hence, the changes in the taxonomic compositions of the microbiota between D7 and D21 did not affect the components and functionalities of these microbiotas that we measured. This agrees with the conclusions from Weimer [47] and Tun et al. [7] that functionalities of gastrointestinal bacteria are shared by different bacterial taxa. This suggests that the taxonomic composition of the microbiota in the various regions of the digestive tract are resilient to a moderate increase in grain feeding, in that their taxonomic composition at the genus level partially returns to the pre-grain feeding state [49].

## 5. Conclusions

Feeding a moderate grain diet for 7 d reduced microbial richness and diversity of microbiota throughout the digestive tract of yearling steers. Extending the duration of this grain feeding up to 21 d did not reduce these richnesses and diversities further. The region in the digestive tract affected the taxonomic composition of the microbiota, as three distinct clusters of microbiota were observed, i.e., the rumen, jejunum/ileum, and the cecum/colon/rectum. Across these regions, grain feeding did not result in a clustering of the taxonomic composition. For all regions, grain feeding did not affect the abundances of the major bacterial phyla, but regions affected these abundances. The relative abundances of genera also differed among regions, with the cecum, colon, and rectum being similar. Grain feeding also affected the relative abundances of genera. Most changes in the relative abundances occurred between 0 and 7 d of grain feeding. Most of the changes in the relative abundances between 7 and 21 d of grain feeding were opposite from the earlier changes, demonstrating the resilience of gastro-intestinal microbiota to moderate increases in grain feeding. This resilience did not apply to the pH and the concentrations of VFA and LPS in the digesta, as these differed between 0 and 7 d of grain feeding, but not between 7 and 21 d of grain feeding. Results show that the duration of moderate grain feeding affects microbiota throughout the digestive tract, but that these effects depend on the duration of this grain feeding.

## Figures and Tables

**Figure 1 microorganisms-08-01854-f001:**
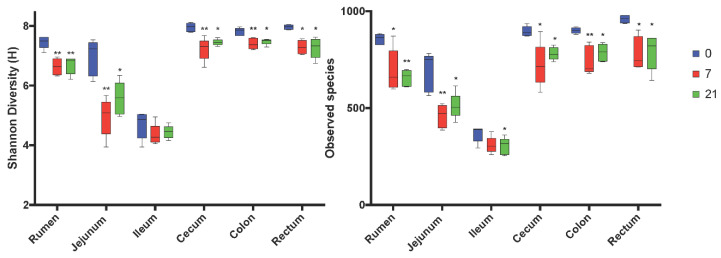
Comparison of Shannon and Observed Number of species alpha-diversity indices by region of the digestive tract and treatment. 0 = no grain feeding, 7 = moderate grain feeding for 7 d, 21 = moderate grain feeding for 21 d. * = significantly (*p*< 0.05) different from D0 by region, ** significantly (*p*< 0.01) different from D0 by region.

**Figure 2 microorganisms-08-01854-f002:**
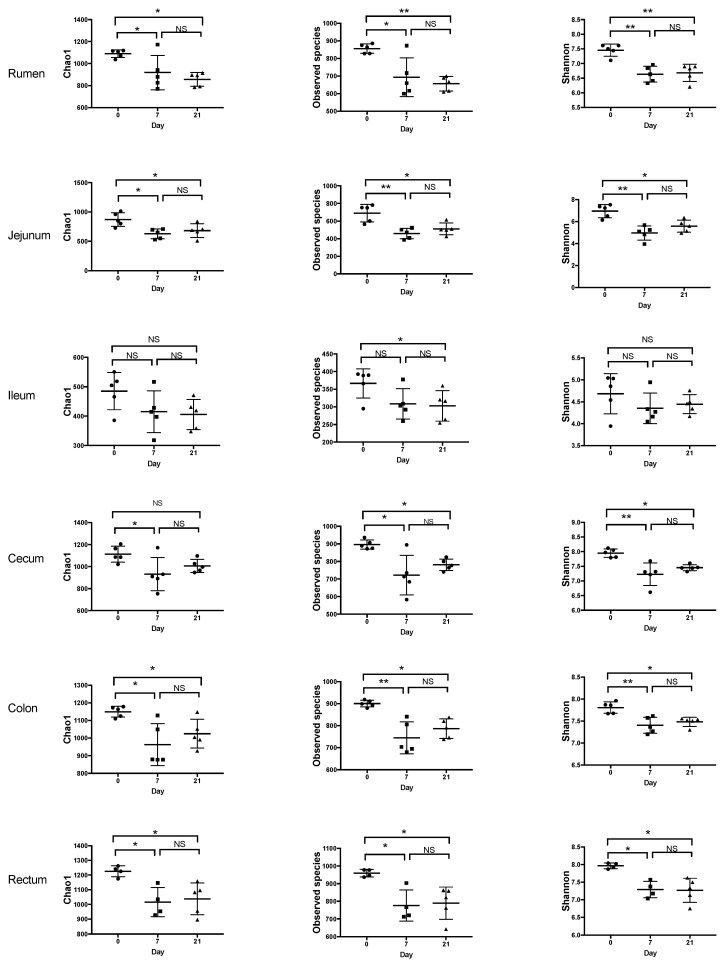
Effect of feeding a moderately-high grain diet on the alpha-diversity of microbiota by region of the digestive tract. Day 0 = no grain feeding, Day 7 = moderate grain feeding for 7 d, Day 21 = moderate grain feeding for 21 d. * = significantly (*p* < 0.05) different, ** = significantly (*p* < 0.01) different, NS = not significantly different.

**Figure 3 microorganisms-08-01854-f003:**
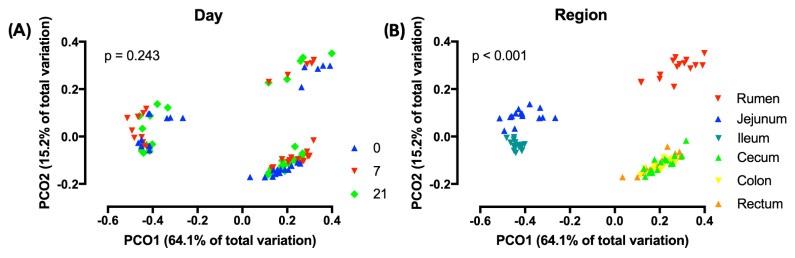
Principal coordinate analyses (PCoA) of weighted UniFrac distances of (**A**) treatments across regions of the digestive tract and (**B**) regions of the digestive tract across treatments. 0 = no grain feeding, 7 = moderate grain feeding for 7 d, 21 = moderate grain feeding for 21 d. Effect of day across regions is not significant (*p* = 0.243) Effect of region across treatment is significant (*p* < 0.001).

**Figure 4 microorganisms-08-01854-f004:**
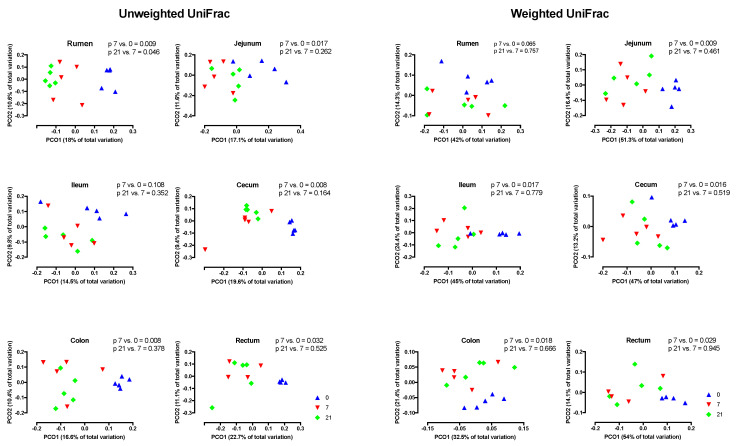
Principal coordinate analyses (PCoA) of unweighted and weighted UniFrac distances of treatments by region of the digestive tract. Day 0 = no grain feeding, Day 7 = moderate grain feeding for 7 d, Day 21 = moderate grain feeding for 21 d.

**Figure 5 microorganisms-08-01854-f005:**
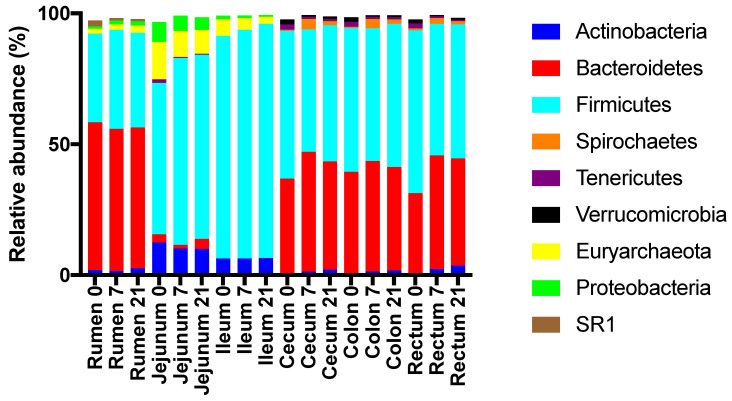
Effect of the duration of feeding a moderate grain diet (0, 7, or 21 d) and region of the digestive tract on the relative abundances of abundant phyla.

**Figure 6 microorganisms-08-01854-f006:**
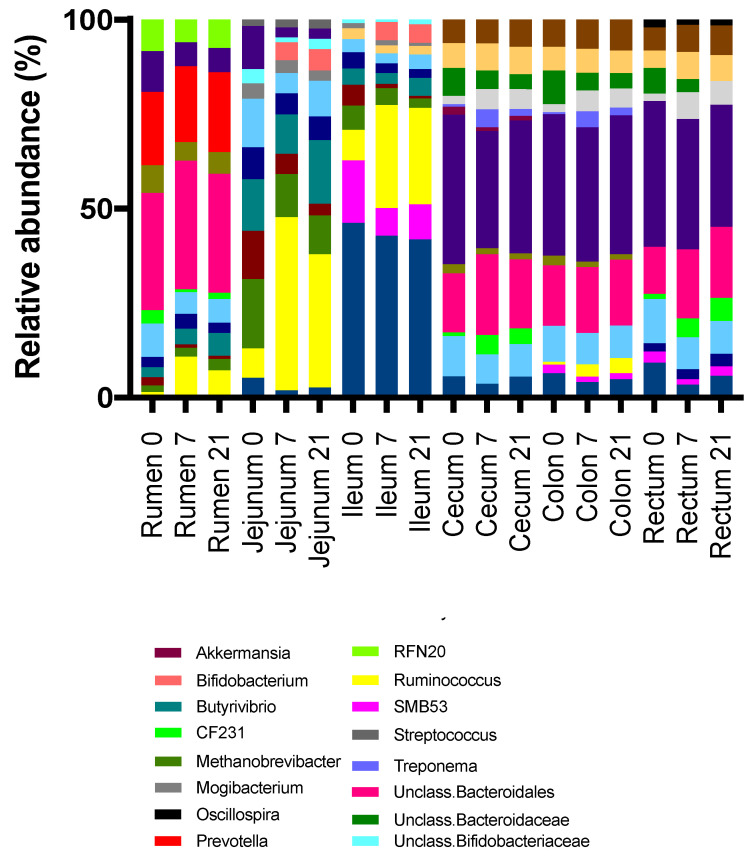
Relative abundances of abundant genera by treatment (0 = no grain feeding, 7 = moderate grain feeding for 7 d, 21 = moderate grain feeding for 21 d) and region of the digestive tract.

**Figure 7 microorganisms-08-01854-f007:**
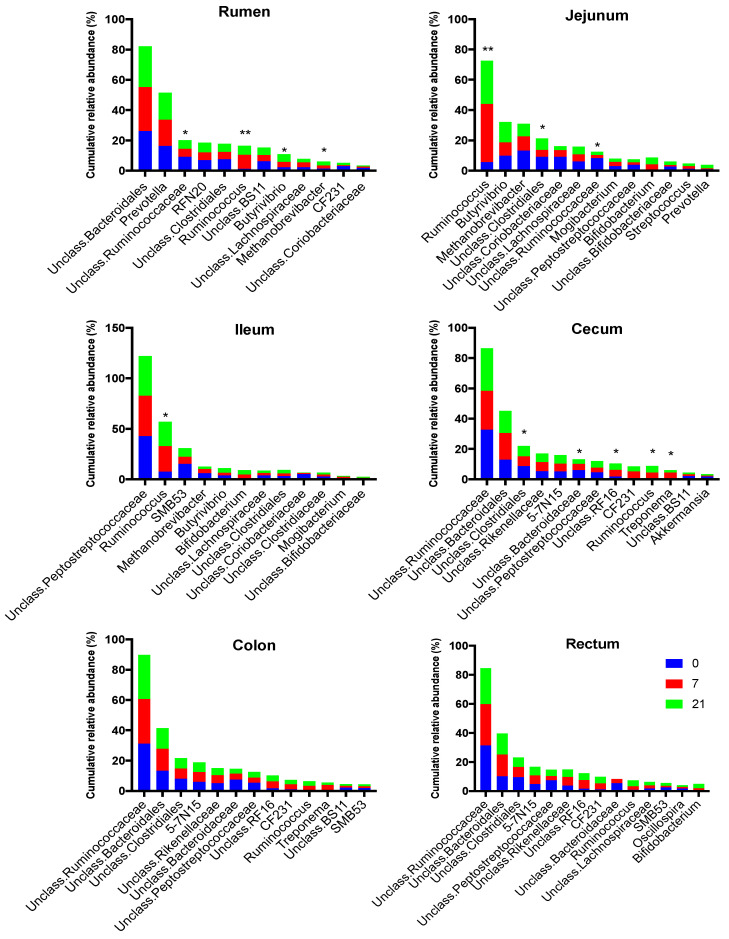
Effect of treatment (0 = no grain feeding, 7 moderate grain feeding for 7 d, 21 = moderate grain feeding for 21 days) on the most abundant genera region of the digestive tract. * = relative abundances of genera differ significantly (*p*< 0.05) among treatments, ** = = relative abundances of genera differ significantly (*p*< 0.01) among treatments

**Figure 8 microorganisms-08-01854-f008:**
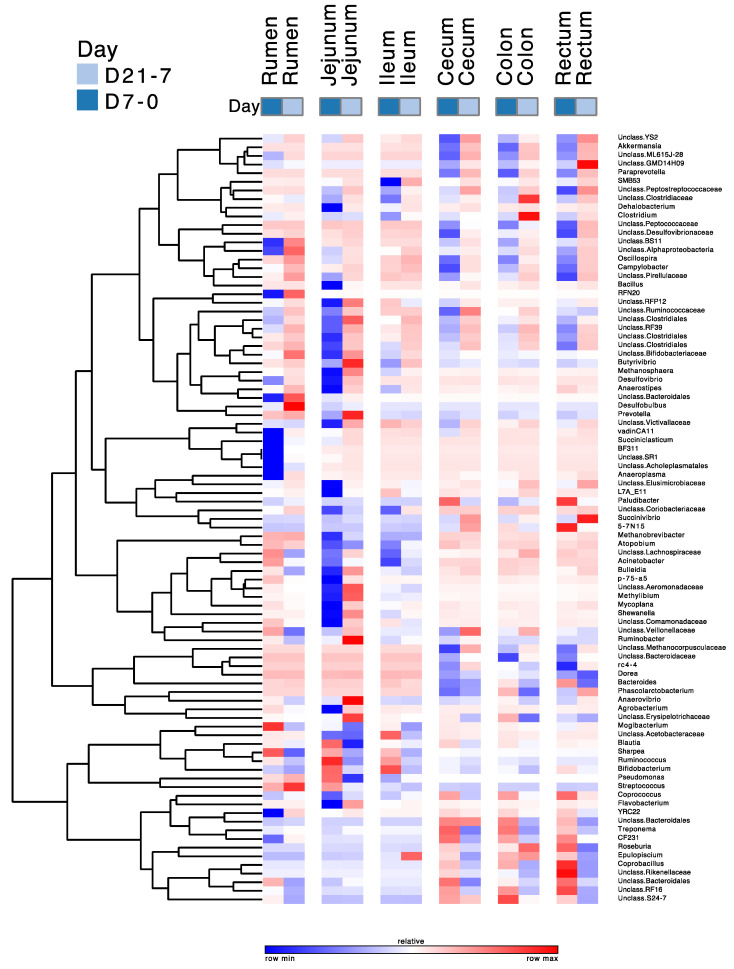
Pattern of change of the relative abundances of abundant genera due to the duration of moderate grain feeding. The magnitude of change varies among lines. D7-0 = 7 d grain feeding vs. no grain feeding; D21-7 = 21 d grain feeding vs. 7 d grain feeding. The scale of the changes varies among rows.

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
