# Peer review of "The Duration of Increased Grain Feeding Affects the Microbiota throughout the Digestive Tract of Yearling Holstein Steers"

_microorganisms, 2020, doi:10.3390/microorganisms8121854_

Round 1

Reviewer 1 Report

Overall, this is a solid study however, I have some issues with it. First, some minor mistakes/typos:

Page 3, line 109

"... described by Derakhshani et al., []." Reference number is missing

Page 3, Lines 122-123

"One sample from the rectum from the D0 treatment was excluded from the analysis because of inadequate sequencing reads." Was this due to the low number of reads or their quality?

Page 4, Lines 162-165

"In the rumen, the relative abundance was highest for D7, whereas this abundance was lowest for D7 in the jejunum, ileum, and colon (Supplementary Table S1). In the rumen, the relative abundance of Bacteroidetes was highest for D7." From the first sentence it is not clear of which phylum or phyla you are referring to?

Beside this small issues, there are some more substantial ones. First part of the Discussion (Lines 257 - 280) is more suited for introduction. Also, there is too much speculation which is not backed up by your results. Essentially you have proven that microbiota clustered by site, which is expected, and that no significant change caused by grain feeding was observed across sites. This is expected. I would suggest you to try modelling microbial dynamics on different levels of taxonomy. This could give you better insight into what happens in each of the sites, because the observed changes in relative abundances are not solely resulting from dietary change (if we hypothesize that this change is not lacking in micro/macro nutrients),  but from the bacterial niche competition (again if we hypothesize all niches are full to begin with) fueled by differences in growth rates and abilities to metabolize different substrates. There is a promising tool for this, you might have heard for: https://github.com/CSB5/BEEM. Also, you are claiming to "may offer the possibility to develop strategies that allow feeding high-grain diets
without disturbing gut health". I would like you to rephrase this, since in truth you are claiming that cattle microbiota can adapt to moderately high grain diet, but you do not offer any palpable strategy.

Author Response

Thank you for your constructive comments. the manuscript was revised and improved accordingly. To address these comments: 

Page 3, line 109

"... described by Derakhshani et al., []." Reference number is missing

AU” Done. See line 120.

Page 3, Lines 122-123 

"One sample from the rectum from the D0 treatment was excluded from the analysis because of inadequate sequencing reads." Was this due to the low number of reads or their quality?

AU: Due to low sequencing reads. See line 134.

Page 4, Lines 162-165

"In the rumen, the relative abundance was highest for D7, whereas this abundance was lowest for D7 in the jejunum, ileum, and colon (Supplementary Table S1). In the rumen, the relative abundance of Bacteroidetes was highest for D7." From the first sentence it is not clear of which phylum or phyla you are referring to?

AU: The first sentence refers to Firmicutes. See line  175.

Beside this small issues, there are some more substantial ones. First part of the Discussion (Lines 257 - 280) is more suited for introduction. Also, there is too much speculation which is not backed up by your results. Essentially you have proven that microbiota clustered by site, which is expected, and that no significant change caused by grain feeding was observed across sites. This is expected. I would suggest you to try modelling microbial dynamics on different levels of taxonomy. This could give you better insight into what happens in each of the sites, because the observed changes in relative abundances are not solely resulting from dietary change (if we hypothesize that this change is not lacking in micro/macro nutrients),  but from the bacterial niche competition (again if we hypothesize all niches are full to begin with) fueled by differences in growth rates and abilities to metabolize different substrates. There is a promising tool for this, you might have heard for: https://github.com/CSB5/BEEM. Also, you are claiming to "may offer the possibility to develop strategies that allow feeding high-grain diets
without disturbing gut health". I would like you to rephrase this, since in truth you are claiming that cattle microbiota can adapt to moderately high grain diet, but you do not offer any palpable strategy.

AU: Lines 58/59 were removed, and, as suggested, a part of the dissuasion was moved to the introduction.  The modelling microbial dynamics on different levels of taxonomy is a great suggestion, but this may have to be done for a follow-up manuscript.  

Reviewer 2 Report

Dear Authors. This article is aimed to topic which is actual permanently – the changes of microbial community after addition of grains to diet of ruminants. This topic is interesting for scientists, but very interesting for farmers owning high producing dairy cows such as Holstein cows, which feeding without grains addition is unthinkable. I emphasize the fact that the description of microbial community was done in throughout whole gastrointestinal tract of steers. On the other hand, I have some comments to your article. From my point of view, there are missing some information and something is described unclear. Manuscript correction according to these comments make this article better readable. For repeating of this experiment some paragraphs in methodology must be corrected. Please see comments below:

Row 66 - ... 5 to 7 mo old ...  better will be month old.

Row 68 - what kind of grass hay was used? Hay from grass monoculture, meadow hay with herbs, hay from grass-trefoil stand or other hay? Different hays have different content of bioactive compounds which can have effect on microbial community in gastro intestinal tract.

Row 68 – the producer of vitamin mineral pellet is missing (Company name, Country)

Row 75 – Is known, that diet structure affects the function of rumen and maybe also the microbial community in GIT of ruminants. The question is: how long were the particles of chopped hay?

Row 80 – Figure 1 show Shannon index and Number of species and not the study design.

Row 64 to 82 – For the repeatability of experiment, all must be exact described. You had 15 animals and 3 sampling points – that means in each sampling time (D0, D7 and D21) you slaughtered 5 animals for samples collecting? Consequently the “n” was 5?

Row 109 – citation Derakhshani et al. () – identification of citation in references is missing.

Row 145 - … the digestive tract with the with the rumen …

Row 162 to 164 – “In the rumen, the relative abundance was highest for D7, whereas this abundance was lowest for D7 in the jejunum, ileum, and colon (Supplementary Table S1).“ – The relative abundance of what was highest? These results are described not clear.

Row 168 and row 219 – there is reference to Figure 8. But, I did not find Figure 8., May be the Figure 9. (row 214) should be numbered with 8.

Author Response

Thank you for your constructive comments. The manuscript was modified and improved accordingly. To address your comments:  

Row 66 - ... 5 to 7 mo old ...  better will be month old.

AU: changed. See line 76.

Row 68 - what kind of grass hay was used? Hay from grass monoculture, meadow hay with herbs, hay from grass-trefoil stand or other hay? Different hays have different content of bioactive compounds which can have effect on microbial community in gastro intestinal tract.

AU: bromegrass hay. See line 66. Composition provided in Supplementary Table 1

Row 68 – the producer of vitamin mineral pellet is missing (Company name, Country)

AU: Company name provide7. See line 69. Composition provided in Supplementary Table 1

Row 75 – Is known, that diet structure affects the function of rumen and maybe also the microbial community in GIT of ruminants. The question is: how long were the particles of chopped hay?

AU: I agree that the particle size of the hay would have had an effect, but unfortunately this size was not measured.   

Row 80 – Figure 1 show Shannon index and Number of species and not the study design.

AU: This sentence was removed.

Row 64 to 82 – For the repeatability of experiment, all must be exact described. You had 15 animals and 3 sampling points – that means in each sampling time (D0, D7 and D21) you slaughtered 5 animals for samples collecting? Consequently the “n” was 5?

AU: This was added. See lines 81.  

Row 109 – citation Derakhshani et al. () – identification of citation in references is missing.

AU: added. See line 120.

Row 145 - … the digestive tract with the with the rumen …

AU: corrected.

Row 162 to 164 – “In the rumen, the relative abundance was highest for D7, whereas this abundance was lowest for D7 in the jejunum, ileum, and colon (Supplementary Table S1).“ – The relative abundance of what was highest? These results are described not clear.

AU: This text was changed by adding that reference was made to Firmicutes. See line 175.

Row 168 and row 219 – there is reference to Figure 8. But, I did not find Figure 8., May be the Figure 9. (row 214) should be numbered with 8.

AU: Yes. Figure 9 should be figure 8. See lines 181 and 226.